**Data Availability Statement:** For this analysis, we used the USAID–DHS program 2016 Ethiopian demographic and health survey data set. To

# Uptake of premarital HIV testing and associated factors among women who had autonomous versus non autonomous marriage in Ethiopia: A nationwide study

Mohammed Ahmed[1]*, Seada Seid[1], Ali Yimer[1], Abdu Seid[2], Ousman Ahmed[3]

1 Department of Public Health, College of Health Science, Woldia University, Woldia, Ethiopia, 2 Department of Midwifery, College of Health Science, Woldia University, Woldia, Ethiopia, 3 Department of Nursing, College of Health Science, Woldia University, Woldia, Ethiopia

* mohaasrar12@gmail.com

## Abstract

### Background

Premarital HIV testing offers an opportunity where prospective couples can know their HIV status before marriage to prevent both heterosexual and vertical transmission of HIV. Therefore, this study aimed to determine whether there is any significant difference in the prevalence of premarital HIV testing among women who had autonomous versus non-autonomous marriage, and to investigate the factors associated with premarital HIV testing among women who had autonomous versus non-autonomous marriage in Ethiopia.

### Methods

Data were extracted from 2016 Ethiopia Demographic and Health Survey dataset and analyzed by using SPSS version 20. Frequencies and weighted percentage of the variables, and second-order Rao-Scott statistic were computed. Multivariable logistic regression analysis was used to identify factors between the two groups. An adjusted odds ratio with 95% confidence interval was considered to state statistically significant associations.

### Result

From 9602 included sample, 4,043 (42.1%) of the women had autonomous marriage, and 5,559(57.9%) of the women had non-autonomous marriage. The prevalence of premarital HIV testing in Ethiopia among women who had autonomous marriage was 30.5% (95% CI: 27.7–33.4%) compared to 20.6% (95% CI: 18.5–22.8) among women who had a non-autonomous marriage. No differences in associated factors were found between women who had autonomous versus non autonomous marriage to uptake HIV testing. In both groups, residence in rural area, education attainment (primary, secondary, higher), media access, being rich and richest, knowing the places for HIV testing, chewing chat, and drinking alcohol were significantly predicts premarital HIV testing.

request the same or different data for another purpose, a new research project request should be submitted to the DHS program here: https://dhsprogram.com/data/dataset/Ethiopia_Standard-DHS_2016.cfm?flag=0. After receiving permission, the researcher can login and select the specific data in the format they prefer.

**Funding:** The author(s) received no specific funding for this work.

**Competing interests:** The authors have declared that no competing interests exist.

## Conclusion

The study indicated that 10% more women in autonomous marriage tested for HIV relative to non-autonomous women whilst being an urban resident, educated, having access to media, household wealth and knowledge of testing facilities significantly predict HIV testing among women in Ethiopia. The paper recommends the Ethiopian government shall expand access to education among women while improving their access to media to enhance their socioeconomic wellbeing and health. Furthermore, it is better to inspire women to undergo autonomous marriage by fostering education in the community to enhance premarital HIV testing.

## Introduction

Women are particularly vulnerable to HIV infection because of increased biological susceptibility to HIV transmission through heterosexual contact [1, 2], and faced a host of structural barriers and contextual gender inequalities [3, 4]. Compared to men, women living in Sub-Saharan Africa (SSA) are more affected by HIV, accounting for 59% of all infections in this region [5]. In Ethiopia, HIV prevalence varies notably by marital status and is 0.8% higher among women who report ever having been married compared with those who have never married (0.3%) [6].

For this reason, HIV testing for marriage applicants is recommended in Africa since it has been acknowledged as a renowned and cost-effective measure [7, 8], as well as a convenient means of HIV infection surveillance [9]. Premarital HIV testing offers an opportunity where prospective couples can know their HIV status before marriage [10] to prevent both heterosexual and vertical transmission of HIV [11–13]. Pieces of evidence showed that about 50–85% of new infections among married/cohabiting partner was due to HIV sero-discordant couples [14, 15], which results in increased risk for HIV negative partners [16].

Recently, many countries including the government of Ethiopia have initiated premarital HIV testing [10, 17], since it is one of the key elements in the prevention and control of HIV/AIDS in the country [10, 18]. According to the 2016 Ethiopia Demographic Health Survey (EDHS) report, 24.5% of married women aged 15–49 ever tested before getting married or living with a partner [19]. A previous study done in Ethiopia among married women showed that being urban residents, attended education, access to media, improved wealth index, known the place of HIV testing, having the discriminatory attitude to a patient living PLHIV, being khat chewer, and alcohol drinker was significantly associated with premarital HIV testing [20]. The above mentioned study merely focused on the factors affecting premarital HIV testing among married women without considering the marital status whether the marriage was autonomous (women who accept marriage proposals on their own volition and without coercion) versus non-autonomous marriage (women whose marriages were decided by partner, families or relatives, and others). Different literature showed that non-autonomous marriage has considerable detrimental health and social consequences such as it exposes them to a lifetime of domestic violence and abuse as they lack standing and power within their households [21], gynecological problems [22], sexually transmitted infections, and psychological and mental disorders [23].

Given this identified gap in the literature, the purpose of this analysis were: 1) To determine whether there is any significant difference in the prevalence of premarital HIV testing among women who had autonomous versus non-autonomous marriage, 2) To investigate the factors associated with premarital HIV testing among women who had autonomous versus non-

autonomous marriage in Ethiopia to tailor specific intervention for the potential delineated factors to hinder HIV transmission.

## Methods and materials

### Data

The current study uses secondary data from the 2016 Ethiopia Demographic Health Survey (EDHS). The 2016 EDHS sample is stratified and was selected in two stages. Each region was stratified into urban and rural areas, which yielded 21 sampling strata. In the first stage, 645 Enumeration areas (EAs) were selected with probability proportional to the EA size and with independent selection in each sampling stratum with the sample allocation. In the second stage of selection, a fixed number of 28 households per cluster were selected with an equal probability systematic selection from the newly created household listing. Thus, a 2016 EDHS cluster is either an EA or a segment of an EA. Based on a fixed sample take of 28 households per cluster, the survey selected 645 EAs, 202 in urban areas and 443 in rural areas. The survey was conducted in 16,650 residential households, A detailed description of the study design and methodology of 2016 EDHS is found in the report [19]. This study was based on the woman questionnaire, which was administered to 15, 683 Ethiopian women aged 15–49 in the selected households.

### Selection criteria

The sample utilized in this study excluded women who were not in a marital union ($n$ = 6,081). The analytic sample for the current study consisted of 9,602 married women. From the included sample, 4,043 of the women had autonomous marriage, and 5,559 of the women had non-autonomous marriage (**Fig 1**).

### Study variables

The dependent variable of the study was a self-reported history of premarital HIV testing among women who had autonomous versus non autonomous marriage. Respondents were asked as "did you undertake an HIV test before you got married?" The responses options were "Yes" or "No".

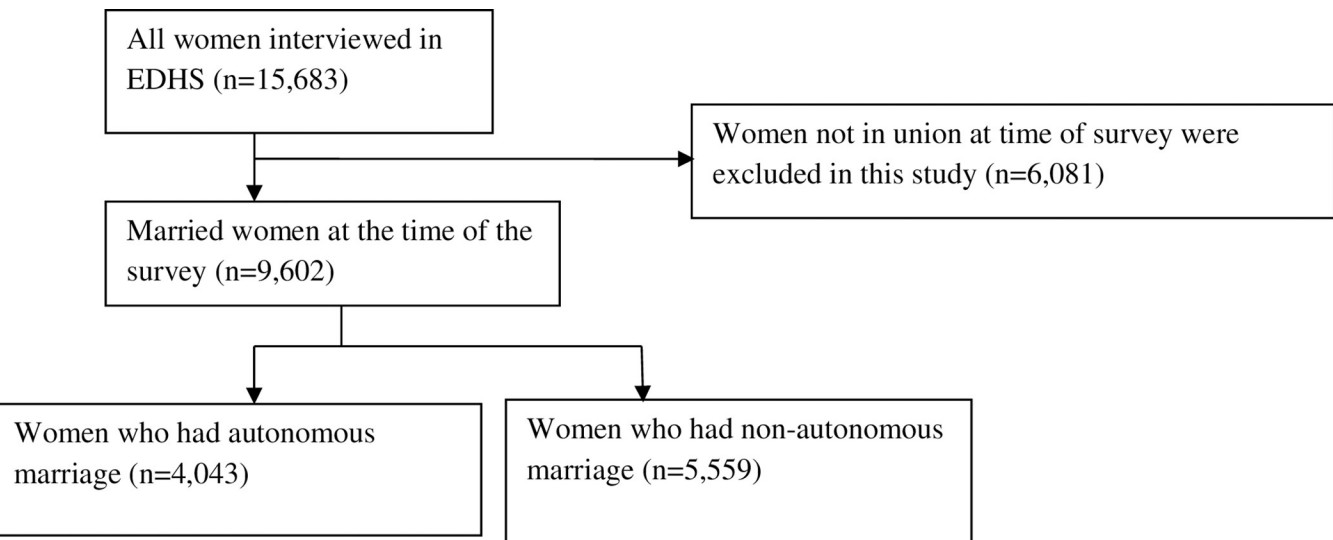

**Fig 1. Flow chart showing the weighted sample used in the study was derived.**

In this study, marriage autonomy was assessed by asking the question in DHS as 'the first time you got married who decide on your marriage?. The questions had the following responses; by myself, parents, other family/ relatives, and others. The responses were coded as women had autonomous marriage if the women decided their marriage by themselves, and women had non-marriage if the women decided their marriage by parents, other family/ relatives, and others.

The independent variables were selected based on a literature review which deemed to be the factors associated with premarital HIV testing and includes age, education status, type of residence, occupation, wealth index (poorest, poorer, middle, richer and richest), media access, knowing the places for HIV testing, comprehensive knowledge about HIV, khat chewing, and alcohol drinking (yes/ no).

*Comprehensive knowledge about HIV/AIDS* was defined based on a widely used measure where each woman was asked whether or not she agreed or disagreed with the following five items: (1) Consistent use of condoms during sexual intercourse can reduce the chance of getting HIV;(2) having just one uninfected faithful partner can reduce the chance of getting HIV; (3) Healthy-looking person can have HIV; (4) HIV can be transmitted by mosquito bites; and (5) a person can become infected by sharing food with a person who has HIV. An additive summary score was created and which was then dichotomized to create a binary variable with 0 indicating at least one incorrect response and 1 to indicate correct response to five items.

*Access to media* was defined based on response to how often respondents read a newspaper, listened to the radio, or watched television. Respondents were asked: "How often do you, read a newspaper, listen to the radio, or watch TV in a week?". Those who responded at least once a week to any of these sources were considered to have access to media/media exposure.

## Statistical analysis

The data were analyzed by using Statistical Package for Social Science (SPSS) version 20. All statistical procedures incorporated complex sampling design analysis applied in the 2016 EDHS. Frequencies and weighted percentage of study variables were calculated. Rao–Scott chi-square test was used to examine the relationship between premarital HIV testing and each of the independent variables separately for women who had autonomous versus non-autonomous marriage. Multivariable binary logistic regression analysis was performed to control confounders and to identify independent factors about premarital HIV testing among each group. All independent variables were entered in the multivariable logistic regression model irrespective of the p-values in the bivariate analysis. Adjusted odds ratio (AOR) with a 95% confidence interval and 2-sided p-value was used to state statistically significant associations. As recommended for complex survey design, sampling weights were applied for this study by dividing the individual women sample weight by 1000,000.

## Ethics approval and consent to participate

Ethical clearance for the study was not required since it was a secondary data analysis from EDHS 2016 database. The researchers had received the survey data from USAID–DHS program and then the researchers of this study have maintained the confidentiality of the data.

## Result

### Socio-demographic characteristics of study participants by type of marriage

From the included sample, 4,043 (42.1%) of the women had autonomous marriage, and 5,559 (57.9%) of the women had non-autonomous marriage. Among women who had autonomous

marriage, the proportion of women who were reside in rural area was 72.8%. Besides, 28% of the study participants were found in the age range of 25–29 years. The participants who didn't attend education shared 44.2%. Moreover, 96.7% of the respondent didn't have comprehensive knowledge about HIV **(Table 1)**.

Similarly, among women who had non-autonomous marriages, the proportion of women who were reside in rural area was 90.4%. Besides, 21.1% of the study subjects were found in the age range of 25–29 years. The participants who didn't attend education shared 71.4%. Moreover, 96.8% of the respondents didn't have comprehensive knowledge about HIV **(Table 2)**.

## Uptake of premarital HIV testing among women who had autonomous versus non autonomous marriage in Ethiopia

Among women who had autonomous marriage, the prevalence of premarital HIV testing was 30.5% (95% CI: 27.7–33.4). Similarly, among women who had non-autonomous marriages, the prevalence of premarital HIV testing was 20.6% (95% CI: 18.5–22.8) (Fig 2).

**Table 1. Characteristics of the study participants who had autonomous marriage (n = 4043).**

| Variables | Category | Overall | Premarital HIV testing among women who had autonomous marriage | | p-value* |
|---|---|---|---|---|---|
| | | | No | Yes | |
| | | n(wt.%) | n (wt. %) | n (wt. %) | |
| Residence | Urban | 1647(27.2) | 714(15.4) | 933(54.0) | P<0.001 |
| | Rural | 2396(72.8) | 1979(84.6) | 417(46.0) | |
| Age | 15–19 | 290(7.2) | 193(7.4) | 97(6.8) | p<0.001 |
| | 20–24 | 877(22.1) | 503(18.6) | 374(29.9) | |
| | 25–29 | 1059(27.9) | 653(26.0) | 406(32.2) | |
| | 30–34 | 746(18.5) | 490(18.5) | 256(18.5) | |
| | 35–39 | 547(12.2) | 417(14.2) | 130(7.6) | |
| | 40–44 | 337(7.6) | 278(9.4) | 59(3.5) | |
| | 45–49 | 187(4.5) | 159(5.9) | 28(1.4) | |
| Educational status | No education | 1731(44.2) | 1554(56.9) | 177(15.3) | p<0.001 |
| | Primary | 1273(35.2) | 769(32.6) | 504(41.0) | |
| | Secondary | 591(11.3) | 245(6.4) | 346(22.3) | |
| | Higher | 448(9.3) | 125(4.1) | 323(21.4) | |
| Occupation | Unemployed | 2314(52.9) | 1701(56.8) | 613(44.0) | p<0.001 |
| | Agricultural | 364(12.8) | 286(14.3) | 78(9.2) | |
| | Non-agricultural | 1365(34.4) | 706(28.9) | 659(46.8) | |
| Access to media | No | 2042(54.7) | 1786(67.1) | 256(26.4) | p<0.001 |
| | Yes | 2001(45.3) | 907(32.9) | 1094(73.6) | |
| Wealth index | Poorest | 937(16.0) | 856(20.6) | 81(5.5) | p<0.001 |
| | Poorer | 468(16.9) | 397(20.7) | 71(8.3) | |
| | Middle | 453(16.7) | 368(19.6) | 85(10.0) | |
| | Richer | 459(17.4) | 338(18.1) | 121(15.9) | |
| | Richest | 1726(33.0) | 734(21.0) | 992(60.4) | |
| Comprehensive knowledge about HIV | No | 3972(96.7) | 2634(96.0) | 1338(98.3) | 0.028 |
| | Yes | 71(3.3) | 59(4.0) | 12(1.7) | |
| Know the places to HIV testing | No | 566(20.8) | 535(29.1) | 31(3.7) | p<0.001 |
| | Yes | 3049(79.2) | 1747(70.9) | 1302(96.3) | |
| Chewing khat | No | 3515(87.1) | 2353(86.2) | 1162(89.1) | |
| | Yes | 528(12.9) | 340(13.8) | 188(10.9) | 0.183 |
| Alcohol drinking | No | 3040(75.1) | 2230(81.1) | 810(61.4) | p<0.001 |
| | Yes | 1003(24.9) | 463(18.9) | 540(38.6) | |

**Table 2. Characteristics of the study participants who didn't have autonomous marriage (n = 5559).**

| Variables | Category | Overall | Premarital HIV testing among women who had not autonomous marriage | | p-value |
|---|---|---|---|---|---|
| | | | No | Yes | |
| | | n(wt.%) | n (wt. %) | n (wt. %) | |
| Residence | Urban | 722(9.6) | 395(6.0) | 327(23.3) | <0.001 |
| | Rural | 4837(90.4) | 4054(94.0) | 783(76.7) | |
| Age | 15–19 | 351(4.8) | 240(4.0) | 111(8.1) | <0.001 |
| | 20–24 | 848(13.5) | 588(11.8) | 260(20.2) | |
| | 25–29 | 1130(21.1) | 817(18.8) | 313(30.3) | |
| | 30–34 | 1068(21.0) | 865(21.4) | 203(19.4) | |
| | 35–39 | 989(17.8) | 872(19.5) | 117(11.6) | |
| | 40–44 | 669(12.0) | 611(13.8) | 58(5.4) | |
| | 45–49 | 504(9.6) | 456(10.8) | 48(5.0) | |
| Educational status | No education | 3894(71.4) | 3394(76.8) | 500(50.6) | <0.001 |
| | Primary | 1348(24.3) | 936(21.4) | 412(35.3) | |
| | Secondary | 248(3.4) | 93(1.6) | 155(10.5) | |
| | Higher | 69(0.9) | 26(0.2) | 43(3.6) | |
| Occupation | Unemployed | 2959(51.5) | 2416(52.2) | 543(48.8) | <0.001 |
| | Agricultural | 1526(29.4) | 1294(30.7) | 232(24.5) | |
| | Non-agricultural | 1074(19.1) | 739(17.1) | 335(26.7) | |
| Access to media | No | 3757(66.2) | 3239(70.8) | 518(48.6) | <0.001 |
| | Yes | 1802(33.8) | 1210(29.2) | 592(51.4) | |
| Wealth index | Poorest | 1943(21.0) | 1742(23.2) | 201(12.4) | <0.001 |
| | Poorer | 1006(22.3) | 851(23.9) | 155(16.4) | |
| | Middle | 889(22.2) | 715(22.7) | 174(20.4) | |
| | Richer | 830(20.8) | 645(20.3) | 185(22.7) | |
| | Richest | 891(13.7) | 496(9.9) | 395(28.1) | |
| Comprehensive knowledge about HIV | No | 5424(96.8) | 4340(96.8) | 1084(96.9) | 0.84 |
| | Yes | 135(3.2) | 109(3.2) | 26(3.1) | |
| Know the places to HIV testing | No | 1306(28.4) | 1254(34.2) | 52(7.4) | p<0.001 |
| | Yes | 3761(71.6) | 2723(65.8) | 1038(92.6) | |
| Chewing khat | No | 4978(84.2) | 3984(84.1) | 994(84.5) | |
| | Yes | 581(15.8) | 465(15.9) | 116(15.5) | 0.89 |
| Alcohol drinking | No | 3611(60.1) | 3002(61.9) | 609(53.2) | 0.005 |
| | Yes | 1948(39.9) | 1447(38.1) | 501(46.8) | |

### Factors associated with premarital HIV testing among women who had autonomous versus non-autonomous marriage in Ethiopia

All the variables were entered into multivariable logistic regression analysis. In both groups, being an urban resident, education attainment (primary, secondary, higher), media access, being rich and richest, knowing the places for HIV testing, chewing chat, and drinking alcohol were significantly associated with premarital HIV testing. No differences in associated factors were found between women who had autonomous versus non autonomous marriage to uptake HIV testing (**Table 3**).

## Discussion

The study aimed to assess uptake of premarital HIV testing and associated factors among women who had autonomous versus non autonomous marriage in Ethiopia. The study finding

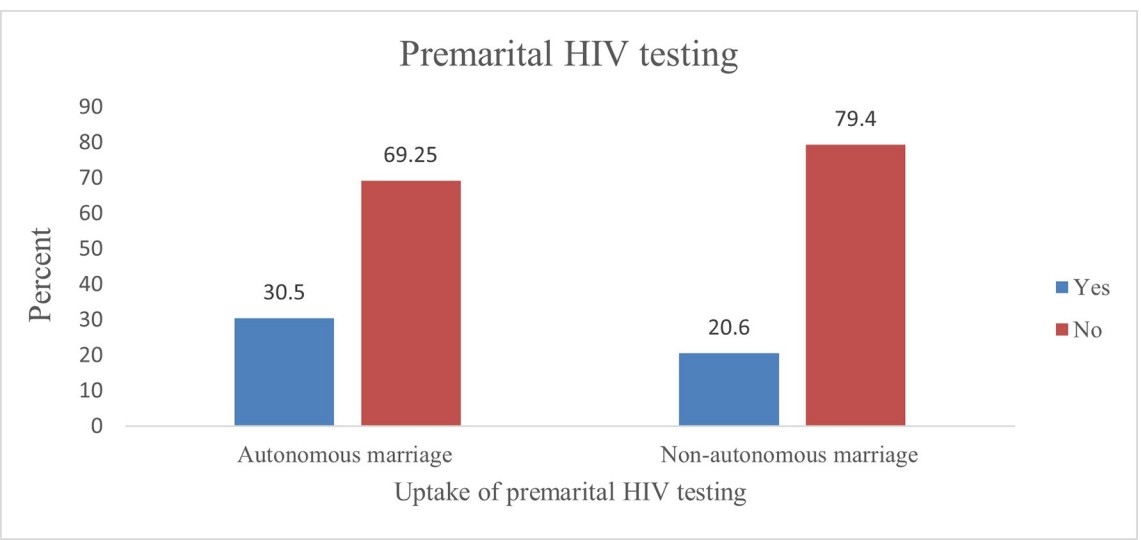

**Fig 2. Uptake of premarital HIV testing among women who had autonomous versus non autonomous marriage in Ethiopia, EDHS 2016.**

showed that 10% more women in autonomous marriage tested for HIV relative to women in non-autonomous marriage. This may be due to the effect of being engaged in autonomous marriage for developing self-rated health. But, this finding also require further studies. Reside in an urban area, being educated, access to media, improved wealth index, knowing the place for HIV testing, chewing chat, and drinking alcohol positively predicts HIV testing in both groups. This showed that no differences in associated factors were appreciated between women who had autonomous versus non autonomous marriage to uptake HIV testing.

Considering residence, women who were residing in urban areas have higher odds to undertake premarital HIV testing. This finding is consistent with a study done in Malawi [24], and Nigeria [25, 26]. The reason for this may be better availability and accessibility of HIV testing facilities in urban settings.

The study further showed that women who were educated have higher odds to carry out premarital HIV testing. This finding is in line with a study conducted in Kenya [27], and Uganda [28]. This could be elucidated by educated women take care of HIV infection, as they easily understood both the transmission and prevention methods [28].

Regarding media access and wealth index, women who had media access and were richer and richest have higher odds to undertake premarital HIV testing. This could be expounded by the possibility that higher income for women enhances their status in the household, enables them to be educated, and can help to have better access to media easily without constraints [28].

The odds of HIV testing were higher among women who knew the place for HIV testing. This finding contrasts with a study conducted in Gambela region, which is found in Ethiopia [29]. This discrepancy may be due to sampling size variation, in which the current study was done based on nationally representative data.

As well, the present study revealed that premarital HIV testing was higher among khat chewers and alcohol drinkers compared to their counterparts. This could be due to risky sexual behavior after alcohol and khat chewing. This may have increased perceived susceptibility to HIV which in turn leads them to be tested for HIV [30, 31].

**Table 3. Multivariable analysis table for identifying factors of premarital HIV testing among women who had autonomous (n = 4043) versus non autonomous (n = 5559) marriage in Ethiopia.**

| Variables | Category | Premarital HIV testing among women who had autonomous marriage | | Premarital HIV testing among women who had not autonomous marriage | |
|---|---|---|---|---|---|
| | | COR(95% CI) | AOR(95% CI) | COR(95% CI) | AOR(95% CI) |
| Residence | Urban | 6.43(4.84–8.53) | 1.94(1.28–2.93)* | 4.72(3.31–6.76) | 1.98(1.18–3.31)* |
| | Rural | 1 | | 1 | 1 |
| Age | 15–19 | 1 | 1 | 1 | 1 |
| | 20–24 | 1.73(1.12–2.68) | 1.01(0.67–1.80) | 0.85(0.59–1.23) | 0.74(0.49–1.12) |
| | 25–29 | 1.34(0.85–2.11) | 0.79(0.49–1.29) | 0.79(0.55–1.15) | 0.72(0.46–1.10) |
| | 30–34 | 1.08(0.66–1.76) | 0.69(0.36–1.33) | 0.45(0.30–0.66) | 0.38(0.24–1.60) |
| | 35–39 | 0.58(0.34–0.97) | 0.37(0.18–1.73) | 0.29(0.19–0.45) | 0.22(0.14–1.35) |
| | 40–44 | 0.41(0.22–0.75) | 0.21(0.09–1.45) | 0.19(0.12–0.31) | 0.14(0.08–1.25) |
| | 45–49 | 0.25(0.11–0.58) | 0.09(0.03–1.32) | 0.23(0.13–0.41) | 0.19(0.09–1.39) |
| Educational status | No education | 1 | 1 | 1 | 1 |
| | Primary | 4.69(3.48–6.33) | 2.25(1.58–3.20)* | 2.51(2.03–3.10) | 1.42(1.12–1.79)* |
| | Secondary | 12.9(8.83–19.1) | 3.19(2.07–4.91)* | 10.2(6.68–15.4) | 2.97(1.83–4.83)* |
| | Higher | 19.5(12.5–30.5) | 4.05(2.38–6.89)* | 22.8(9.71–53.7) | 5.82(2.38–14.2)* |
| Occupation | Unemployed | 1 | 1 | 1 | 1 |
| | Agricultural | 0.82(0.56–1.21) | 1.34(0.86–2.07) | 0.85(0.67–1.08) | 0.85(0.65–1.11) |
| | Non-agricultural | 2.08(1.61–2.70) | 1.01(0.74–1.35) | 1.67(1.27–2.17) | 0.98(0.74–1.30) |
| Access to media | No | 1 | 1 | 1 | 1 |
| | Yes | 5.70(4.45–7.30) | 1.52(1.06–2.16)* | 2.56(2.07–3.19) | 1.42(1.11–1.81)* |
| Wealth index | Poorest | 1 | 1 | 1 | 1 |
| | Poorer | 1.51(0.88–2.58) | 1.11(0.64–1.91) | 1.27(0.94–1.73) | 1.04(0.73–1.47) |
| | Middle | 1.93(1.17–3.17) | 1.07(0.65–1.79) | 1.67(1.18–2.36) | 1.37(0.94–2.01) |
| | Richer | 3.32(2.06–5.37) | 1.53(0.90–2.61) | 2.07(1.51–2.86) | 1.57(1.08–2.28)* |
| | Richest | 10.8(7.02–16.7) | 1.87(1.01–3.46)* | 5.25(3.67–7.51) | 1.71(1.10–2.66)* |
| Comprehensive knowledge about HIV | No | 1 | 1 | 1 | 1 |
| | Yes | 0.43(0.19–0.93) | 0.86(0.44–1.69) | 0.94(0.54–1.66) | 1.64(0.93–2.92) |
| Know the places to HIV testing | No | 1 | 1 | 1 | 1 |
| | Yes | 10.7(6.23–18.7) | 4.87(2.69–8.79)* | 6.48(4.08–10.3) | 5.55(3.49–8.85)* |
| Chewing chat | No | 1 | 1 | 1 | 1 |
| | Yes | 0.76(0.51–1.14) | 1.21(1.12–1.88)* | 1.25(1.03–1.53)* | 1.75(1.08–2.85)* |
| Alcohol drinking | No | 1 | 1 | 1 | 1 |
| | Yes | 2.69(2.03–3.58) | 1.52(1.12–2.06)* | 1.43(1.12–1.84) | 1.56(1.19–2.02)* |

*Statistically significant at a p-value of <0.05.

## Strength and limitation of the study

Although findings in this study are useful for policy, there are some noteworthy limitations. Since, the data were extracted from secondary data, the issue of under or over reporting of the study outcomes may be evident. Notwithstanding these limitations, this is the only study done using a nationally representative dataset to assess the uptake of premarital HIV testing and its factors among women who had autonomous versus non autonomous marriage in Ethiopia up to date.

## Conclusions

The paper shows that 10% more women in autonomous marriage tested for HIV relative to non-autonomous women. No differences in associated factors were found between women

who had autonomous versus non autonomous marriage to uptake HIV testing. Being an urban resident, educated, having access to media, household wealth and knowledge of testing facilities significantly predict HIV testing among women in Ethiopia. The paper recommends that the Ethiopian government shall encourage autonomous marriage, expand access to education among women while improving their access to media to enhance their socioeconomic wellbeing and health to augment premarital HIV testing. Future researches should focus on the effect of autonomous marriage on health care service utilization on a strong study design.

## Acknowledgments

We are grateful to the USAID–DHS program for providing access to 2016 Ethiopian Demographic Health Survey

## Author Contributions

**Conceptualization:** Mohammed Ahmed, Seada Seid, Ali Yimer, Abdu Seid.

**Data curation:** Mohammed Ahmed.

**Formal analysis:** Mohammed Ahmed.

**Methodology:** Mohammed Ahmed.

**Software:** Mohammed Ahmed.

**Visualization:** Mohammed Ahmed.

**Writing – original draft:** Mohammed Ahmed, Seada Seid, Abdu Seid.

**Writing – review & editing:** Mohammed Ahmed, Seada Seid, Ali Yimer, Abdu Seid, Ousman Ahmed.

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
