## [Decision Letter · Decision Letter 0]

29 Sep 2021

PONE-D-21-07892

Does the magnitude of premarital HIV testing differ among women who had autonomous versus nonautonomous marriage in Ethiopia? Evidence from a nationwide household survey

PLOS ONE

Dear Dr. Ahmed,

Thank you for submitting your manuscript to PLOS ONE. After careful consideration, we feel that it has merit but does not fully meet PLOS ONE’s publication criteria as it currently stands. Therefore, we invite you to submit a revised version of the manuscript that addresses the points raised during the review process.

The four reviewers have identified a number of concerns that need to be carefully addressed in a revision to your manuscript.

We look forward to receiving your revised manuscript.

Kind regards,

Jamie Males

Staff Editor

PLOS ONE

Journal Requirements:

2. Thank you for submitting the above manuscript to PLOS ONE. During our internal evaluation of the manuscript, we found significant text overlap between your submission and the following previously published works, some of which you are an author.

https://journals.plos.org/plosone/article?id=10.1371%2Fjournal.pone.0235830

Please revise the manuscript to rephrase the duplicated text, cite your sources, and provide details as to how the current manuscript advances on previous work. Please note that further consideration is dependent on the submission of a manuscript that addresses these concerns about the overlap in text with published work.

Reviewers' comments:

Reviewer's Responses to Questions

**Comments to the Author**

1. Is the manuscript technically sound, and do the data support the conclusions?

Reviewer #1: Yes

Reviewer #2: Partly

Reviewer #3: Yes

Reviewer #4: Yes

2. Has the statistical analysis been performed appropriately and rigorously? 

Reviewer #1: Yes

Reviewer #2: No

Reviewer #3: Yes

Reviewer #4: Yes

3. Have the authors made all data underlying the findings in their manuscript fully available?

Reviewer #1: Yes

Reviewer #2: Yes

Reviewer #3: Yes

Reviewer #4: Yes

4. Is the manuscript presented in an intelligible fashion and written in standard English?

Reviewer #1: No

Reviewer #2: No

Reviewer #3: Yes

Reviewer #4: No

5. Review Comments to the Author

Reviewer #1: General Comment

The topic under study is of public health importance in sub-Saharan Africa, as forced marriages are rampant. I find the findings from this study, therefore, relevant in informing public health practice especially premarital HIV testing strategies. I have made comments/suggestions to improve the manuscript.

Comment #1

Title: The study assessed the magnitude and associated factors of premarital HIV testing among women who had autonomous versus non-autonomous marriage in Ethiopia. It is not appropriate to phrase the title as a question for this study, yet this question was answered by their findings. The authors should revise this. I suggest they revise to something close to this: “Magnitude and associated factors of premarital HIV testing among women in autonomous and non-autonomous marriages in Ethiopia: a nationwide survey”

Comment #2

Abstract: Some conclusions on associated factors are lacking results. Please include the results for the objective of associated factors.

Comment #4

Results: The statement “About 72.8% of the women were rural residents” should be revised, because this is the exact percent in table 1, so “about” does not apply. This should be revised throughout the results section.

Comment #5

Results: Do not start sentences with numbers written in figures (E.g., 96.8%), please revise this throughout.

Comment #6

Change the word “predictors” to .. factors associated, or associated factors” the results, since this was a cross-sectional study.

Comment #7

Discussion: Paragraph one, the authors should not repeat results with the confidence intervals, since this was already highlighted in the results section. Instead they should give a short narrative of the same results. Please revise.

Comment #8

Discussion: Major comment-Authors should add a paragraph on the implications of the study findings.

Comment #9

Table 1 & 2: Revise the table titles. I suggest you refrain from use of univariate and bivariate in the same title, as this can be confusing. Clearly these tables shows characteristics of the study participants among women in autonomous and non-autonomous marriages, by premarital HIV testing status

-Include denominators in column headings for the two columns under premarital HIV testing.

Reviewer #2: Does the magnitude of premarital HIV testing differ among women who had autonomous versus nonautonomous marriage in Ethiopia? Evidence from a nationwide household survey

Reviewer’s report

I’d like to thank the editors for giving me the opportunity to review the manuscript and I congratulate the authors for their work. The authors used data from a nationally representative survey to determine the extent to which women were tested for HIV prior to marriage in Ethiopia. I doubt whether the paper can be published in its current form. Some sections of the manuscript may not be clear to potential readers. I’d suggest that the authors provide their revised manuscript to someone with strong English Language proficiency to read the manuscript and correct some typing and grammatical errors prior to resubmission. The following comments and suggestions could help improve the quality of manuscript:

INTRODUCTION

Paragraph 3: The authors cited a study conducted in Ethiopia. What was the target group for the cited study? Was it for married women? Please specify.

Paragraph 4: The authors should define autonomous and non-autonomous marriage to their readers. If the definition is provided at this stage, they don’t need to repeat the definition in the subsequent sections of the manuscript.

Paragraph 5: Based on my review of this manuscript and comments in the findings/results section (see below), I suggest that the authors reformulate the objectives of their analysis. The following could serve as a guide: The objectives of this analysis were: 1) To estimate the prevalence of pre-marital HIV testing (PMHT) among married women in Ethiopia; 2) To determine whether there is any significant difference in the prevalence of PMHT among women who had autonomous vs non-autonomous marriage, 3) To investigate the factors associated with PMHT among married women in Ethiopia.

Since data were derived from a nationally representative population-based survey, I suggest that authors use prevalence rather than magnitude. In the last sentence of the introduction section, the authors should briefly outline the relevance and usefulness of their study findings.

METHODS and material section

While we understand that the detailed methods are found in the EHDS report, the authors didn’t outline how cluster sampling technique was conducted. They simply stated that: “The first and second stages involved the selection of clusters and households in each cluster, respectively” This is too brief. The authors should outline (briefly) how the first stage and second stage sampling was conducted then reference the EDHS report for more details. In the last sentence of paragraph 1 (Methods section), the authors should indicate the number of households that participated in the EDHS.

Selection criteria: If the authors define the different types of marriage in the Introduction section, they don’t need to repeat it here. I suggest that the definition in brackets should be deleted.

Study Variables

The authors did not explain how their dependent variable (pre-marital HIV testing) was assessed in the survey. Please kindly include this in your manuscript so that readers can understand how the variable was evaluated. Furthermore, the explanation of how marriage was classified as autonomous and non-autonomous is not clearly elaborated. The authors should rewrite the sentence.

Independent variables: The authors should indicate how the variables: alcohol drinking and wealth index were assessed in the survey.

It is not clear how the authors determined women with or without comprehensive knowledge of HIV status. Please kindly explain how this was done.

Statistical Analysis

Was the data set used in this analysis already weighted by the EDHS team before being provided to the authors or it was unweighted data? If unweighted, the authors should provide details of the sampling design analysis so that readers should understand how this was done. The authors simply stated that “All statistical procedures incorporated complex sampling design analysis applied in the 2016 EDHS.” In my view, this statement doesn’t explain to the readers how this was done.

The authors should conduct Chi-square test to determine whether there were differences in demographic characteristics among the 2 groups of married women.

The authors indicated that …” All independent variables were entered in the multivariable logistic regression model irrespective of the p-values in the bivariate analysis.” Could you explain to your readers the essence of conducting a bivariate association despite the fact that you still included all independent variables in the model. Did the authors test for multicollinearity prior to multivariate analysis? If not, why? What was your level of statistical significance? Where the tests conducted 2-sided? Please provide details to your readers.

FINDINGS

The findings of this analysis shows that the factors associated with pre-marital HIV testing (PMHT) among women with autonomous marriage were the same for women with non-autonomous marriage. Consequently, I think there was no added value examining and presenting these same factors in different tables for both groups of women. This could be considered as repeating the analysis without any new information and perhaps explains why there are lots of data/tables in the manuscript.

If the analysis had showed that the factors were substantially different among the 2 sub-groups, then it would have been interesting to conduct a sub-group analysis and report the results.

As a result, I suggest that the authors reanalyze the pooled data (combining both groups) and investigate the factors associated with PMHT among married women in Ethiopia. The authors main comparison should focus on investigating whether there was a significant difference in the prevalence of PMHT among women who had autonomous vs non-autonomous marriages. If there is a difference, the authors should provide an explanation (in the Discussion Section) for the observed results taking into consideration the socio-cultural context of Ethiopia.

Discussion

The authors cited studies conducted in Kenya, Uganda, Malawi, and Nigeria to compare and contrast the findings of their study. What was the target group for these studies? Was the target group married women or women with autonomous/non-autonomous marriages. It is appropriate that the authors cite studies conducted among the same target population while comparing and contrasting their findings. Please consider this as you revise your manuscript.

Limitations

The authors outline only one limitation of their analysis. Furthermore, the statement on “difficult to find causal relationships due to contemporaneous ……”is not clear to their readers. Please provide details so that it should be clear to your readers.

What about limitations associated with the fact that responses for PMHT were self-reported? Could the participants have under-reported or over-reported that they received HIV testing before marriage? Was there any limitation associated with secondary data used in the analysis? Please provide details of the limitations of the analysis. What do the authors recommend for future studies in other settings?

In what ways do you think the findings of this analysis/study could help inform HIV testing policies and guidelines in Ethiopia and beyond especially with regards to PMHT.

Reviewer #3: General remarks:

Thank you for putting together this interesting study. I believe the findings of this study will be very valuable in informing interventions for HIV testing in Ethiopia among the two groups.

Title:

It’s rhetorical to ask a question only to answer it yourself. I would be prefer you state your title in simple but informative manner.

I think that the use of uptake of premarital HIV testing would be better than magnitude of premarital HIV testing

Abstract:

The background does not underscore the study gap. This should come out clearly to inform your study aim.

The result section should report on the demographics first (how many women were in the category of autonomous versus nonautomonous marriage) before you report on the other objectives. Include predictors as part of the results section.

The conclusion should be based on the key research question (HIV testing) and any recommendation as provided. 1-3 sentences is more than enough for a conclusion.

Introduction:

What do you mean by marriage applicants in paragraph 2?

Some of your sentences overflow. My suggestion would be that a good sentence is <40 words (1-3 lines). Anything more than that should be revised so that you write with clarity.

I like that last paragraph that clearly underscores the knowledge gap.

Methods and Materials

You may need to go through the PLOS ONE guidelines so that you have clear sections for your method section e.g. study design and setting; sample size, data management and analysis; ethics and approval.

Where the independent variables part of the dataset.

Why did you opt to use Rao-Scott Chi-Square?

Multivariate analysis usually involve statistically significant variables at bivariate analysis of sometimes independent variables <0.2. Why did you choose to include every variable irrespective of p-value?

Results:

I think the order of reporting is paramount. The first part should always involve the demographic information; part two-outcome variable; part three-predictors.

A figure e.g. bar graph would clearly show the comparison for your outcome variable.

It’s advisable to very short sentences that start with figures (numbers).

In the first paragraph for the predictors, you should tell us the significant variables at chi-square test while quoting their p-value, e.g. age (p<0.001). Then in the second paragraph you report on the variables independently associated with HIV testing while quoting their AOR, 95% CI, and p-value. E.g age (AOR: 1.34; 95% CI: 1.22-2.30; p<0.001).

We don’t interpret statistics at results level, we do that during the discussion so it would be advisable to avoid statements like marriage were 1.94 times higher etc.

Could you clarify for me your outcome variable?

Discussion

You have a lot of data that you could discuss, but you need to be deliberate on those which are interesting and worth focusing on. I like the first paragraph you have put forward which summarizes the key findings of your study but you could make it better and more focused.

Have clear paragraphs for your discussion. A good paragraph ranges from 3-6 sentences in 6-12 lines (average 8 lines).

For each of the discussion paragraph, you should: a) Reecho your findings (interpret the results), b) explain your findings c) compare and contrast findings with previous literature d) Justify differences in findings if any e) state the public health significance of your findings

Conclusion

Have a sentence which makes suggestions for area(s) of future research

Table 1 and 2:

You could remove columns for cross-tabulations and simply report on p-value

Reviewer #4: The paper seeks to assess the prevalence and associated factors that influence premarital HIV testing among women who are in autonomous versus those in non-autonomous marriages. The focus on women is necessitated by the vulnerability of women to HIV infection primarily due to their biological make up. Using the 2016 Ethiopian Demographic and Health Survey and applying descriptive and multivariate logistic regression to the data, the paper shows that 10% more women in autonomous marriage (30.5%) tested for HIV relative to non-autonomous women (20.6%) whilst being an urban resident, educated, having access to media, household wealth and knowledge of testing facilities significantly predict HIV testing among women in Ethiopia. The paper recommends that the Ethiopian government should expand access to education among women while improving their access to media to enhance their socioeconomic wellbeing and health. Overall, the methods employed for the data analysis-descriptive statistics and multivariate logistic regression are appropriate and in line with the study's objectives.

The contribution of this paper to the literature is minuscule given the plethora of studies on the phenomenon in both developing and developed countries. The paper’s novelty however lies in its focus on women in union and disaggregation of the analysis by women in autonomous versus non autonomous marriage. However, the paper does not clearly articulate the distinction between autonomous and non-autonomous marriage, which is key to the analysis. For instance, while the ‘definition’ of autonomous marriage is quite straightforward (women who accept marriage proposals on their own volition and without coercion) that of non-autonomous (women whose marriages were decided by partner, families or relatives, and others) is a bit confusing Does it mean that some partners forced marriage on their partners? Or better still some family members forced some women to marry their partners against their will. The paper assumes that the 9,602 women in the sample had married customarily. However, reasonable number of women marries by ordnance in Ethiopia, and such women are more likely to check their status compared to women who are ‘forced’ to marry their partners against their will.

The authors should also take note of the following suggestions in revising their manuscripts;

• The use of ‘magnitude’ as in the proportion of the sampled women who had autonomous marriage compared to those who had non-autonomous marriage in the text is somehow misleading. The correct term should be proportion or percentage or better still prevalence.

• The research gap been addressed did not come out clearly. It will be helpful if the authors could review additional literature to help clearly articulate the research gap being filled.

• Before disaggregating the data into women in autonomous marriage versus non-autonomous marriage, it will be insightful to estimate a pooled multivariate logistic model consisting of both samples. In this case, you can introduce a dummy for either autonomous or non-autonomous marriage to ascertain its effect of HIV testing status.

• The discussion of the findings could improve by providing theoretical and intuitive explanations for the findings while comparing them to related findings in the extant literature.

Decision

I recommend a major revision but I leave the final verdict to the editors.

6. PLOS authors have the option to publish the peer review history of their article (what does this mean?). If published, this will include your full peer review and any attached files.

Reviewer #1: No

Reviewer #2: No

Reviewer #3: No

Reviewer #4: **Yes: **Edward Nketiah-Amponsah

---

## [Author Response · Author response to Decision Letter 0]

15 Feb 2022

Thank you very much for PLOS one editorial office, academic editors, as well as reviewers of this manuscript entitled Does the magnitude of premarital HIV testing differ among women who had autonomous versus non autonomous marriage in Ethiopia? Evidence from a nationwide household survey for their astonished effort.

The written documents below explained point by point response for respective reviewer’s comment.

Reviewer 1 comments and authors response : 

Reviewer #1: General Comment

The topic under study is of public health importance in sub-Saharan Africa, as forced marriages are rampant. I find the findings from this study, therefore, relevant in informing public health practice especially premarital HIV testing strategies. I have made comments/suggestions to improve the manuscript.

Comment #1

Title: The study assessed the magnitude and associated factors of premarital HIV testing among women who had autonomous versus non-autonomous marriage in Ethiopia. It is not appropriate to phrase the title as a question for this study, yet this question was answered by their findings. The authors should revise this. I suggest they revise to something close to this: “Magnitude and associated factors of premarital HIV testing among women in autonomous and non-autonomous marriages in Ethiopia: a nationwide survey”

Author response: The authors amended the manuscript based on the reviewer comments 

Comment #2

Abstract: Some conclusions on associated factors are lacking results. Please include the results for the objective of associated factors.

Author response: The authors amended the manuscript based on the reviewer comments 

Comment #4

Results: The statement “About 72.8% of the women were rural residents” should be revised, because this is the exact percent in table 1, so “about” does not apply. This should be revised throughout the results section.

Author response: The authors amended the manuscript based on the reviewer comments.

Comment #5

Results: Do not start sentences with numbers written in figures (E.g., 96.8%), please revise this throughout.

Author response: The authors amended the manuscript based on the reviewer comments.

Comment #6

Change the word “predictors” to .. Factors associated, or associated factors” the results, since this was a cross-sectional study.

Author response: The authors amended the manuscript based on the reviewer comments.

Author response: 

Comment #7

Discussion: Paragraph one, the authors should not repeat results with the confidence intervals, since this was already highlighted in the results section. Instead they should give a short narrative of the same results. Please revise.

Author response: The authors amended the manuscript based on the reviewer comments.

Comment #8

Discussion: Major comment-Authors should add a paragraph on the implications of the study findings.

Comment #9

Table 1 & 2: Revise the table titles. I suggest you refrain from use of univariate and bivariate in the same title, as this can be confusing. Clearly these tables shows characteristics of the study participants among women in autonomous and non-autonomous marriages, by premarital HIV testing status

-Include denominators in column headings for the two columns under premarital HIV testing.

Author response: The authors amended the manuscript based on the reviewer comments.

 

Reviewer #2 comments and author response: 

Comment 1: INTRODUCTION

Paragraph 3: The authors cited a study conducted in Ethiopia. What was the target group for the cited study? Was it for married women? Please specify.

Author response: the cited study was merely focused among married women without considering the marital status whether the marriage was autonomous (women who accept marriage proposals on their own volition and without coercion) versus non-autonomous marriage (women whose marriages were decided by partner, families or relatives, and others).

Comment 2: Paragraph 4: The authors should define autonomous and non-autonomous marriage to their readers. If the definition is provided at this stage, they don’t need to repeat the definition in the subsequent sections of the manuscript.

Author response: The authors amended the manuscript based on the reviewer comments.

Comment 3: Paragraph 5: Based on my review of this manuscript and comments in the findings/results section (see below), I suggest that the authors reformulate the objectives of their analysis. The following could serve as a guide: The objectives of this analysis were: 1) To estimate the prevalence of pre-marital HIV testing (PMHT) among married women in Ethiopia; 2) To determine whether there is any significant difference in the prevalence of PMHT among women who had autonomous vs non-autonomous marriage, 3) To investigate the factors associated with PMHT among married women in Ethiopia.

Since data were derived from a nationally representative population-based survey, I suggest that authors use prevalence rather than magnitude. In the last sentence of the introduction section, the authors should briefly outline the relevance and usefulness of their study findings.

Author response: The authors amended the manuscript based on the reviewer comments.

Comment 4: METHODS and material section

While we understand that the detailed methods are found in the EHDS report, the authors didn’t outline how cluster sampling technique was conducted. They simply stated that: “The first and second stages involved the selection of clusters and households in each cluster, respectively” This is too brief. The authors should outline (briefly) how the first stage and second stage sampling was conducted then reference the EDHS report for more details. In the last sentence of paragraph 1 (Methods section), the authors should indicate the number of households that participated in the EDHS.

Author response: The authors amended the manuscript based on the reviewer comments and included in the revised manuscript method section. 

Comment 5: Selection criteria: If the authors define the different types of marriage in the Introduction section, they don’t need to repeat it here. I suggest that the definition in brackets should be deleted.

Author response: The authors amended the manuscript based on the reviewer comments and included in the revised manuscript.

Comment 6: Study Variables

The authors did not explain how their dependent variable (pre-marital HIV testing) was assessed in the survey. Please kindly include this in your manuscript so that readers can understand how the variable was evaluated. Furthermore, the explanation of how marriage was classified as autonomous and non-autonomous is not clearly elaborated. The authors should rewrite the sentence.

Author response: The authors amended the manuscript based on the reviewer comments, and included in the revised manuscript. The dependent variable of the study was a self-reported history of premarital HIV testing among women who had autonomous versus non autonomous marriage (yes or no response). In this study, marriage autonomy was assessed by asking the question in DHS as ‘the first time you got married who decide on your marriage?. The questions had the following responses; by myself, parents, other family/ relatives, and others. The responses were coded as women had autonomous marriage if the women decided their marriage by themselves, and women had non-marriage if the women decided their marriage by parents, other family/ relatives, and others.

Comment 7: Independent variables: The authors should indicate how the variables: alcohol drinking and wealth index were assessed in the survey. It is not clear how the authors determined women with or without comprehensive knowledge of HIV status. Please kindly explain how this was done.

Author response: already explained in the study variable section like this Comprehensive knowledge about HIV/AIDS was defined based on a widely used measure where each woman was asked whether or not she agreed or disagreed with the following five items: (1) Consistent use of condoms during sexual intercourse can reduce the chance of getting HIV;(2) having just one uninfected faithful partner can reduce the chance of getting HIV; (3) Healthy-looking person can have HIV; (4) HIV can be transmitted by mosquito bites; and (5) a person can become infected by sharing food with a person who has HIV. An additive summary score was created and which was then dichotomized to create a binary variable with 0 indicating at least one incorrect response and 1 to indicate correct response to five items. 

Alcohol drinking was assessed by yes/no response, Wealth index was assessed by PCA and categorized as poorest, poorer, middle, richer , richest 

Comment 8: Statistical Analysis

Was the data set used in this analysis already weighted by the EDHS team before being provided to the authors or it was unweighted data? If unweighted, the authors should provide details of the sampling design analysis so that readers should understand how this was done. The authors simply stated that “All statistical procedures incorporated complex sampling design analysis applied in the 2016 EDHS.” In my view, this statement doesn’t explain to the readers how this was done.

The authors should conduct Chi-square test to determine whether there were differences in demographic characteristics among the 2 groups of married women.

The authors indicated that …” All independent variables were entered in the multivariable logistic regression model irrespective of the p-values in the bivariate analysis.” Could you explain to your readers the essence of conducting a bivariate association despite the fact that you still included all independent variables in the model. Did the authors test for multicollinearity prior to multivariate analysis? If not, why? What was your level of statistical significance? Where the tests conducted 2-sided? Please provide details to your readers.

Author response: I already conduct chi-square test (Rao-scot second order statistic) to determine whether there were differences in demographic characteristics among the 2 groups of married women, and to identify candidates for multivariable model. All the variables were entered in multivariable model since they have clinical significance for the study. The level of statistical significance was 2- sided p-value with 95% confidence interval. As recommended for complex survey design, sampling weights were applied for this study by dividing the individual women sample weight by 1000,000.

Comment 9: FINDINGS

The findings of this analysis shows that the factors associated with pre-marital HIV testing (PMHT) among women with autonomous marriage were the same for women with non-autonomous marriage. Consequently, I think there was no added value examining and presenting these same factors in different tables for both groups of women. This could be considered as repeating the analysis without any new information and perhaps explains why there are lots of data/tables in the manuscript. If the analysis had showed that the factors were substantially different among the 2 sub-groups, then it would have been interesting to conduct a sub-group analysis and report the results. As a result, I suggest that the authors re analyze the pooled data (combining both groups) and investigate the factors associated with PMHT among married women in Ethiopia. The authors main comparison should focus on investigating whether there was a significant difference in the prevalence of PMHT among women who had autonomous vs non-autonomous marriages. If there is a difference, the authors should provide an explanation (in the Discussion Section) for the observed results taking into consideration the socio-cultural context of Ethiopia. 

Author response: Factors affecting premarital HIV testing among married women in Ethiopia is already done previously, but the current study focus the prevalence of PMHT and its associated factors independently among the two groups to appreciate the differences.

Comment 10: Discussion

The authors cited studies conducted in Kenya, Uganda, Malawi, and Nigeria to compare and contrast the findings of their study. What was the target group for these studies? Was the target group married women or women with autonomous/non-autonomous marriages. It is appropriate that the authors cite studies conducted among the same target population while comparing and contrasting their findings. Please consider this as you revise your manuscript.

Author response: The authors amended the manuscript based on the reviewer comments and included in the revised manuscript

Comment 11: Limitations

The authors outline only one limitation of their analysis. Furthermore, the statement on “difficult to find causal relationships due to contemporaneous ……”is not clear to their readers. Please provide details so that it should be clear to your readers.

What about limitations associated with the fact that responses for PMHT were self-reported? Could the participants have under-reported or over-reported that they received HIV testing before marriage? Was there any limitation associated with secondary data used in the analysis? Please provide details of the limitations of the analysis. What do the authors recommend for future studies in other settings?

In what ways do you think the findings of this analysis/study could help inform HIV testing policies and guidelines in Ethiopia and beyond especially with regards to PMHT.

Author response: The authors amended the manuscript based on the reviewer comments and included in the revised manuscript in strength and limitation of the study like this ‘’although findings in this study are useful for policy, there are some noteworthy limitations. For example, the data were extracted from secondary data and our study may not be devoid of the shortcomings associated with this approach. Besides, since it was based on self-reporting the issue of under or over reporting of the study outcomes may be evident. Notwithstanding these limitations, this is the only study done using a nationally representative dataset to assess the uptake of premarital HIV testing and its factors among women who had autonomous versus non autonomous marriage in Ethiopia up to date.’’ 

Reviewer #3: General remarks:

Thank you for putting together this interesting study. I believe the findings of this study will be very valuable in informing interventions for HIV testing in Ethiopia among the two groups.

Title:

It’s rhetorical to ask a question only to answer it yourself. I would be prefer you state your title in simple but informative manner.

Comment 1: I think that the use of uptake of premarital HIV testing would be better than magnitude of premarital HIV testing

Author response: The authors amended the manuscript based on the reviewer comments and included in the revised manuscript

Comment 2: Abstract:

The background does not underscore the study gap. This should come out clearly to inform your study aim.

The result section should report on the demographics first (how many women were in the category of autonomous versus nonautomonous marriage) before you report on the other objectives. Include predictors as part of the results section.

The conclusion should be based on the key research question (HIV testing) and any recommendation as provided. 1-3 sentences is more than enough for a conclusion.

Author response: The authors amended the manuscript based on the reviewer comments and included in the revised manuscript

Comment 3: Introduction:

What do you mean by marriage applicants in paragraph 2?

Author response: Marriage applicant’s means female or male who wants to engage in marriage 

Comment 4: Some of your sentences overflow. My suggestion would be that a good sentence is <40 words (1-3 lines). Anything more than that should be revised so that you write with clarity.

I like that last paragraph that clearly underscores the knowledge gap.

Author response: The authors amended the manuscript based on the reviewer comments and included in the revised manuscript

Comment 5: Methods and Materials

You may need to go through the PLOS ONE guidelines so that you have clear sections for your method section e.g. study design and setting; sample size, data management and analysis; ethics and approval.

Where the independent variables part of the dataset.

Why did you opt to use Rao-Scott Chi-Square?

Multivariate analysis usually involve statistically significant variables at bivariate analysis of sometimes independent variables <0.2. Why did you choose to include every variable irrespective of p-value?

Author response: the author prepared the revised manuscript based on PLos guideline. I already conduct chi-square test (Rao-scot second order statistic) to determine whether there were differences in demographic characteristics among the 2 groups of married women, and to identify candidates for multivariable model. All the variables were entered in multivariable model since they have clinical significance for the study. The level of statistical significance was 2- sided p-value with 95% confidence interval.

Comments 6: Results:

I think the order of reporting is paramount. The first part should always involve the demographic information; part two-outcome variable; part three-predictors.

A figure e.g. bar graph would clearly show the comparison for your outcome variable.

It’s advisable to very short sentences that start with figures (numbers).

Author response: The authors amended the manuscript based on the reviewer comments and included in the revised manuscript.

In the first paragraph for the predictors, you should tell us the significant variables at chi-square test while quoting their p-value, e.g. age (p<0.001). Then in the second paragraph you report on the variables independently associated with HIV testing while quoting their AOR, 95% CI, and p-value. E.g age (AOR: 1.34; 95% CI: 1.22-2.30; p<0.001).

Author response: The second paragraph showed the multivariable model 

We don’t interpret statistics at results level, we do that during the discussion so it would be advisable to avoid statements like marriage were 1.94 times higher etc.

Author response: The authors amended the manuscript based on the reviewer comments and included in the revised manuscript.

Could you clarify for me your outcome variable?

Author response: The outcome variable is self-reporting history of premarital HIV testing.

Comment 7: Discussion

You have a lot of data that you could discuss, but you need to be deliberate on those which are interesting and worth focusing on. I like the first paragraph you have put forward which summarizes the key findings of your study but you could make it better and more focused.

Have clear paragraphs for your discussion. A good paragraph ranges from 3-6 sentences in 6-12 lines (average 8 lines).

For each of the discussion paragraph, you should: a) Reecho your findings (interpret the results), b) explain your findings c) compare and contrast findings with previous literature d) Justify differences in findings if any e) state the public health significance of your findings

Author response: The authors amended the manuscript based on the reviewer comments and included in the revised manuscript.

Conclusion

Have a sentence which makes suggestions for area(s) of future research

Author response: The authors amended the manuscript based on the reviewer comments and included in the revised manuscript.

Reviewer #4 comments and author response: 

Comment 1: The authors should also take note of the following suggestions in revising their manuscripts;

• The use of ‘magnitude’ as in the proportion of the sampled women who had autonomous marriage compared to those who had non-autonomous marriage in the text is somehow misleading. The correct term should be proportion or percentage or better still prevalence.

Author response: The authors amended the manuscript based on the reviewer comments, and included in the revised manuscript

Comment 2: The research gap been addressed did not come out clearly. It will be helpful if the authors could review additional literature to help clearly articulate the research gap being filled.

Author response: The authors amended the manuscript based on the reviewer comments, and included in the revised manuscript

Comment 3: Before disaggregating the data into women in autonomous marriage versus non-autonomous marriage, it will be insightful to estimate a pooled multivariate logistic model consisting of both samples. In this case, you can introduce a dummy for either autonomous or non-autonomous marriage to ascertain its effect of HIV testing status.

Author response: factors affecting premarital HIV testing among married women in Ethiopia is already done previously, but the current study focus the prevalence of PMHT and its associated factors independently among the two groups to appreciate the differences. 

Comment 4: The discussion of the findings could improve by providing theoretical and intuitive explanations for the findings while comparing them to related findings in the extant literature.

Author response: The authors amended the manuscript based on the reviewer comments, and included in the revised manuscript

Best regards///

---

## [Decision Letter · Decision Letter 1]

30 Mar 2022

PONE-D-21-07892R1Uptake of premarital HIV testing and its associated factors among women who had autonomous versus non autonomous marriage in Ethiopia: a nationwide studyPLOS ONE

Dear Dr. Ahmed,

Thank you for submitting your manuscript to PLOS ONE. After careful consideration, we feel that it has merit but does not fully meet PLOS ONE’s publication criteria as it currently stands. Therefore, we invite you to submit a revised version of the manuscript that addresses the points raised during the review process.

The manuscript has been evaluated by three reviewers, and their comments are available below.

The reviewers have raised a number of concerns that need attention. They request additional information on methodological aspects of the study and results, as well as requested for English language copyediting/grammar, and additional questions.

Could you please revise the manuscript to carefully address the concerns raised?

We look forward to receiving your revised manuscript.

Kind regards,

Sebastian Shepherd

Associate Editor

PLOS ONE

Journal Requirements:

Reviewers' comments:

Reviewer's Responses to Questions

**Comments to the Author**

1. If the authors have adequately addressed your comments raised in a previous round of review and you feel that this manuscript is now acceptable for publication, you may indicate that here to bypass the “Comments to the Author” section, enter your conflict of interest statement in the “Confidential to Editor” section, and submit your "Accept" recommendation.

Reviewer #1: All comments have been addressed

Reviewer #2: (No Response)

Reviewer #3: (No Response)

2. Is the manuscript technically sound, and do the data support the conclusions?

Reviewer #1: Yes

Reviewer #2: Yes

Reviewer #3: Yes

3. Has the statistical analysis been performed appropriately and rigorously? 

Reviewer #1: Yes

Reviewer #2: No

Reviewer #3: Yes

4. Have the authors made all data underlying the findings in their manuscript fully available?

Reviewer #1: Yes

Reviewer #2: Yes

Reviewer #3: Yes

5. Is the manuscript presented in an intelligible fashion and written in standard English?

Reviewer #1: Yes

Reviewer #2: No

Reviewer #3: No

6. Review Comments to the Author

Reviewer #1: I thank the authors for addressing my comments.

However, they should do a final grammar check; for instance, the first sentence of the abstract has a mixing of past and present tenses. The authors should change the word "faced" to "face".

Similarly, the ethics approval statement should be stated in the past tense, not the present tense.

Reviewer #2: I have reviewed for the second time the manuscript titled: “Uptake of premarital HIV testing and its associated factors among women who had autonomous versus non autonomous marriage in Ethiopia: a nationwide study.” While the authors have responded to the initial comments, it is my view that this paper can not be published in its current form. There are several typos and grammatical errors which need to be corrected. Sharing the manuscript to an English native speaker for review and correction of these errors will be very helpful. Furthermore, there are other methodological aspects that could be addressed to improve the quality of the manuscript.

Comments:

Page 1

Line 4. For the sake of coherence, I suggest that the 3rd sentence in the first paragraph (“In Ethiopia….” ) should appear immediately after the 1st sentence.

Page 1

Paragraph 3: Correct the typo by using capitalization. It should be written as Ethiopia Demographic Health Survey”, not Ethiopia demographic health Survey (EDHA). There are many similar typos and grammatical errors in the manuscript that should be corrected.

Methods section:

In the selection criteria, the authors should insert a comma for numbers which are more than 999. Hence, “9602” should be written as “9,602”. Please carefully review the manuscript and correct all similar typos including those in Figure 1.

I think it would be helpful to the readers if the authors insert the % of women for each subgroup. In the sentence, “ From the included sample, 4043 of the women had autonomous marriage, and 5559 of the women had non-autonomous marriage” could be written as “From the included sample, 4,043 (42.1%) of the women had autonomous marriage, and 5,559(57.9%) of the women had non-autonomous marriage”.

Study variables:

The authors should include the question that was asked to evaluate whether women undertook premarital HIV testing before marriage. A statement such as: [Respondents were asked “Did you undertake an HIV test before you got married?” The responses options were “Yes” or “No”]. This is an example, the authors should correct the manuscript using the exact question that was asked in the survey.

Access to media: The authors should indicate the response options that were used in the survey. For example, respondents were asked: “How often do you, read a newspaper, listen to the radio, or watch TV in a week?” What were the other response options aside from “once a week”? The authors should provide adequate information, so that researchers in other countries can replicate the study.

Results

I suggest that instead of the title: “Descriptive statistics of women who had autonomous versus non-autonomous marriage”, the authors should say “Socio-demographic characteristics of study participants by type of marriage”

Under the subheading: “Uptake of premarital HIV testing among women who had autonomous versus non autonomous marriage in Ethiopia”, the authors write: “Among 4043 women who had autonomous marriage, the magnitude of premarital HIV testing”. I suggest that the authors replace the word “magnitude” with “prevalence” as previously suggested.

This analysis shows that there wasn’t any relevance of conducting a subgroup analysis of the factors associated with premarital HIV among autonomous vs non-autonomous marriage because the factors identified are the same for both groups. What additional value does this add? If the government of a country has enacted a policy that women should undertake HIV testing before marriage, the decision to take the test is not influenced by the type of marriage (autonomous vs non-autonomous marriage.) Therefore, whether the marriage is autonomous vs non-autonomous, women are expected to undergo HIV testing.

It is this reviewer’s view that the central focus of this article which unfortunately the authors did not take advantage was to estimate the % of women who underwent HIV testing before marriage, determine whether there was any statistically significant difference in the % of women who received HIV testing before marriage among the 2 groups and then examine the factors associated with premarital HIV testing among married women in Ethiopia in a pooled multivariate logistic model consisting of both samples. I will therefore suggest that the authors conduct a pooled analysis.

In the discussion section, the authors compared and contrasted their findings with studies that examined premarital HIV status among married women. There was very little or no discussion on autonomous vs non-autonomous marriage, which demonstrates that that there wasn’t any new or interesting finding from the subgroup analysis (autonomous and non-autonomous marriage.)

The argument the authors are making is that the factors affecting premarital HIV testing among married women in Ethiopia is known, but that their study was focused on estimating the prevalence of premarital HIV testing and its associated factors independently among the two groups to appreciate the differences. Unfortunately, the factors were the same, which suggest that there no value of separating them.

If the subgroup analysis generated different results/factors associated with…., that would have justified their approach.

Reviewer #3: General remarks:

I thank the authors for their efforts to address the comments. The manuscript is now in better shape than the first version.

Title

The title is clear and informative.

I suggest the authors remove the word, ‘its’ just before associated factors

Abstract

Background:

The background does not underscore the study gap. This should come out clearly to inform your study aim.

Method:

Good

Results:

The result section should report on the demographics first (how many women were in the category of autonomous versus nonautomonous marriage) before you report on the other objectives. Include predictors as part of the results section.

The last sentence is missing a word (of)

Do you mean residence in a rural area?

Conclusion:

The opening words are redundant….The study concludes that

Numbers ten and above should be written in Arabic numeral e.g. 10% and not in words (ten percent)

The paper recommends that the……is also redundant

Please use active voice while writing your abstract

Introduction

The introduction section is clear and much focused except for some grammatical issues.

I suggest the authors enlist the support of a native English to proofread the paper or English editing software.

I prefer you state the purpose of the study instead of the objectives in the last paragraph.

Methods and Materials

Give a reference for your data analysis software, SPSS.

The operational definitions of knowledge about HIV and media access could be entered as new paragraphs without bolding the words

Results

The second, third, and fourth sentences of the Results section have major grammatical issues that need to be addressed.

We use the terminology study participants NOT study subject.

Avoid the phrase, ‘same fashion’. Scholarly writing demands scholar words and or phrases

I like figure but it needs to be well formatted with the same word style and font as the entire document. Also, remove the longitudinal lines.

Since the p-values are not included in Table 3, you could quote the p-values in your description of the significant factors in the text.

Discussion

I am not comfortable with the opening paragraph of your discussion. You could start this section by saying…..We aimed to assess the prevalence of HIV testing and associated factors among……. We found that…..

Then, you go ahead and pick each component of this summary and discuss following the earlier guidelines

For each of the discussion paragraphs, you should: a) State your findings (interpret the results), b) explain your findings c) compare and contrast findings with previous literature d) Justify differences in findings if any e) state the public health significance of your findings

The second sentence under strengths and limitations is not clear. Please revise this.

Conclusion

Have a sentence that makes suggestions for the area(s) of future research

Also, make a recommendation for autonomous marriage.

7. PLOS authors have the option to publish the peer review history of their article (what does this mean?). If published, this will include your full peer review and any attached files.

Reviewer #1: No

Reviewer #2: No

Reviewer #3: No

---

## [Author Response · Author response to Decision Letter 1]

14 Apr 2022

Thank you very much for PLOS one editorial office, academic editors, as well as reviewers of this manuscript entitled uptake of premarital HIV testing and associated factors among women who had autonomous versus non autonomous marriage in Ethiopia: a nationwide study.

The written documents below explained point by point response for respective reviewer’s comment.

Reviewer 1 comments and authors response: 

Reviewer #1: I thank the authors for addressing my comments.

However, they should do a final grammar check; for instance, the first sentence of the abstract has a mixing of past and present tenses. The authors should change the word "faced" to "face".

Similarly, the ethics approval statement should be stated in the past tense, not the present tense.

Author response: All the comments are addressed in the submitted revised manuscript.

Reviewer 2 comments and authors response: 

Reviewer #2: I have reviewed for the second time the manuscript titled: “Uptake of premarital HIV testing and its associated factors among women who had autonomous versus non autonomous marriage in Ethiopia: a nationwide study.” While the authors have responded to the initial comments, it is my view that this paper can not be published in its current form. There are several typos and grammatical errors which need to be corrected. Sharing the manuscript to an English native speaker for review and correction of these errors will be very helpful. Furthermore, there are other methodological aspects that could be addressed to improve the quality of the manuscript.

Comments:

Page 1

Line 4. For the sake of coherence, I suggest that the 3rd sentence in the first paragraph (“In Ethiopia….” ) Should appear immediately after the 1st sentence.

Author response: The authors amended the manuscript based on the reviewers comment. 

Page 1

Paragraph 3: Correct the typo by using capitalization. It should be written as Ethiopia Demographic Health Survey”, not Ethiopia demographic health Survey (EDHA). There are many similar typos and grammatical errors in the manuscript that should be corrected.

Author response: The authors amended the manuscript based on the reviewers comment.

Methods section:

In the selection criteria, the authors should insert a comma for numbers which are more than 999. Hence, “9602” should be written as “9,602”. Please carefully review the manuscript and correct all similar typos including those in Figure 1.

Author response: The authors amended the manuscript based on the reviewers comment.

I think it would be helpful to the readers if the authors insert the % of women for each subgroup. In the sentence, “ From the included sample, 4043 of the women had autonomous marriage, and 5559 of the women had non-autonomous marriage” could be written as “From the included sample, 4,043 (42.1%) of the women had autonomous marriage, and 5,559(57.9%) of the women had non-autonomous marriage”.

Author response: The authors amended the manuscript based on the reviewers comment. 

Study variables:

The authors should include the question that was asked to evaluate whether women undertook premarital HIV testing before marriage. A statement such as: [Respondents were asked “Did you undertake an HIV test before you got married?” The responses options were “Yes” or “No”]. This is an example, the authors should correct the manuscript using the exact question that was asked in the survey.

Author response: The authors amended the manuscript based on the reviewers comment.

Access to media: The authors should indicate the response options that were used in the survey. For example, respondents were asked: “How often do you, read a newspaper, listen to the radio, or watch TV in a week?” What were the other response options aside from “once a week”? The authors should provide adequate information, so that researchers in other countries can replicate the study.

Author response: The authors amended the manuscript based on the reviewers comment.

Results

I suggest that instead of the title: “Descriptive statistics of women who had autonomous versus non-autonomous marriage”, the authors should say “Socio-demographic characteristics of study participants by type of marriage”

Author response: The authors amended the manuscript based on the reviewers comment.

Under the subheading: “Uptake of premarital HIV testing among women who had autonomous versus non autonomous marriage in Ethiopia”, the authors write: “Among 4043 women who had autonomous marriage, the magnitude of premarital HIV testing”. I suggest that the authors replace the word “magnitude” with “prevalence” as previously suggested.

Author response: The authors amended the manuscript based on the reviewers comment.

This analysis shows that there wasn’t any relevance of conducting a subgroup analysis of the factors associated with premarital HIV among autonomous vs non-autonomous marriage because the factors identified are the same for both groups. What additional value does this add? If the government of a country has enacted a policy that women should undertake HIV testing before marriage, the decision to take the test is not influenced by the type of marriage (autonomous vs non-autonomous marriage.) Therefore, whether the marriage is autonomous vs non-autonomous, women are expected to undergo HIV testing.

It is this reviewer’s view that the central focus of this article which unfortunately the authors did not take advantage was to estimate the % of women who underwent HIV testing before marriage, determine whether there was any statistically significant difference in the % of women who received HIV testing before marriage among the 2 groups and then examine the factors associated with premarital HIV testing among married women in Ethiopia in a pooled multivariate logistic model consisting of both samples. I will therefore suggest that the authors conduct a pooled analysis.

Author response: The intention of this paper is to see whether the factors are different or not between the two groups. The pooled analysis is already done in the previous published research paper. Even if the factors among the sub groups are the same, the strength of the association is different. 

In the discussion section, the authors compared and contrasted their findings with studies that examined premarital HIV status among married women. There was very little or no discussion on autonomous vs non-autonomous marriage, which demonstrates that that there wasn’t any new or interesting finding from the subgroup analysis (autonomous and non-autonomous marriage.)

Author response: The authors amended the manuscript based on the reviewers comment.

Reviewer 3 comments and authors response:

Reviewer #3: General remarks:

I thank the authors for their efforts to address the comments. The manuscript is now in better shape than the first version.

Title

The title is clear and informative.

I suggest the authors remove the word, ‘its’ just before associated factors

Author response: The authors amended the manuscript based on the reviewers comment.

Abstract

Background:

The background does not underscore the study gap. This should come out clearly to inform your study aim.

Author response: The authors amended the manuscript based on the reviewers comment.

Method:

Good

Results:

The result section should report on the demographics first (how many women were in the category of autonomous versus nonautomonous marriage) before you report on the other objectives. Include predictors as part of the results section.

The last sentence is missing a word (of)

Do you mean residence in a rural area?

Author response: The authors amended the manuscript based on the reviewers comment.

Conclusion:

The opening words are redundant….The study concludes that

Numbers ten and above should be written in Arabic numeral e.g. 10% and not in words (ten percent)

The paper recommends that the……is also redundant

Please use active voice while writing your abstract

Author response: The authors amended the manuscript based on the reviewers comment.

Introduction

The introduction section is clear and much focused except for some grammatical issues.

I suggest the authors enlist the support of a native English to proofread the paper or English editing software.

I prefer you state the purpose of the study instead of the objectives in the last paragraph.

Author response: The authors amended the manuscript based on the reviewers comment.

Methods and Materials

Give a reference for your data analysis software, SPSS.

The operational definitions of knowledge about HIV and media access could be entered as new paragraphs without bolding the words

Author response: The authors amended the manuscript based on the reviewers comment.

Results

The second, third, and fourth sentences of the Results section have major grammatical issues that need to be addressed.

We use the terminology study participants NOT study subject.

Avoid the phrase, ‘same fashion’. Scholarly writing demands scholar words and or phrases

I like figure but it needs to be well formatted with the same word style and font as the entire document. Also, remove the longitudinal lines.

Since the p-values are not included in Table 3, you could quote the p-values in your description of the significant factors in the text.

Author response: The authors amended the manuscript based on the reviewers comment. 

Discussion

I am not comfortable with the opening paragraph of your discussion. You could start this section by saying…..We aimed to assess the prevalence of HIV testing and associated factors among……. We found that…..

Then, you go ahead and pick each component of this summary and discuss following the earlier guidelines

For each of the discussion paragraphs, you should: a) State your findings (interpret the results), b) explain your findings c) compare and contrast findings with previous literature d) Justify differences in findings if any e) state the public health significance of your findings

The second sentence under strengths and limitations is not clear. Please revise this.

Author response: The authors amended the manuscript based on the reviewers comment.

Conclusion

Have a sentence that makes suggestions for the area(s) of future research

Also, make a recommendation for autonomous marriage.

Author response: The authors amended the manuscript based on the reviewers comment. 

Thank you for reviewer’s constructive comments

---

## [Decision Letter · Decision Letter 2]

16 Jun 2022

PONE-D-21-07892R2Uptake of premarital HIV testing and  associated factors among women who had autonomous versus non autonomous marriage in Ethiopia: a nationwide studyPLOS ONE

Dear Dr. Ahmed,

Thank you for submitting your manuscript to PLOS ONE. After careful consideration, we feel that it has merit but does not fully meet PLOS ONE’s publication criteria as it currently stands. Therefore, we invite you to submit a revised version of the manuscript that addresses the points raised during the review process.

Specifically in the Results section please define whether you mean factors associated with premarital HIV testing among women who had autonomous versus non-autonomous marriage in Ethiopia.

We look forward to receiving your revised manuscript.

Kind regards,

Colin Johnson, Ph.D.

Academic Editor

PLOS ONE

Journal Requirements:

Reviewers' comments:

Reviewer's Responses to Questions

**Comments to the Author**

1. If the authors have adequately addressed your comments raised in a previous round of review and you feel that this manuscript is now acceptable for publication, you may indicate that here to bypass the “Comments to the Author” section, enter your conflict of interest statement in the “Confidential to Editor” section, and submit your "Accept" recommendation.

Reviewer #2: (No Response)

Reviewer #3: All comments have been addressed

2. Is the manuscript technically sound, and do the data support the conclusions?

Reviewer #2: (No Response)

Reviewer #3: Yes

3. Has the statistical analysis been performed appropriately and rigorously? 

Reviewer #2: (No Response)

Reviewer #3: Yes

4. Have the authors made all data underlying the findings in their manuscript fully available?

Reviewer #2: (No Response)

Reviewer #3: Yes

5. Is the manuscript presented in an intelligible fashion and written in standard English?

Reviewer #2: (No Response)

Reviewer #3: Yes

6. Review Comments to the Author

Reviewer #2: (No Response)

Reviewer #3: Results

Do you mean factors associated with premarital HIV testing among women who had autonomous versus

non-autonomous marriage in Ethiopia?

Please revise this sub-heading

7. PLOS authors have the option to publish the peer review history of their article (what does this mean?). If published, this will include your full peer review and any attached files.

Reviewer #2: No

Reviewer #3: No

---

## [Author Response · Author response to Decision Letter 2]

22 Jun 2022

Thank you very much for PLOS one editorial office, academic editors, as well as reviewers of this manuscript entitled uptake of premarital HIV testing and associated factors among women who had autonomous versus non autonomous marriage in Ethiopia: a nationwide study.

The written documents below explained point by point response for respective reviewer and editor comments.

Reviewer 1 comments and authors response: 

Please define whether you mean factors associated with premarital HIV testing among women who had autonomous versus non-autonomous marriage in Ethiopia.

Author response: I have included in the revised manuscript 

Editor comments and author response

Author response: I have reviewed the references in the manuscript and I did not found retracted articles. 

 Thank you for reviewers and editor for their constructive comments

---

## [Editor Report · Decision Letter 3]

11 Jul 2022

Uptake of premarital HIV testing and  associated factors among women who had autonomous versus non autonomous marriage in Ethiopia: a nationwide study

PONE-D-21-07892R3

Dear Dr. Ahmed,

We’re pleased to inform you that your manuscript has been judged scientifically suitable for publication and will be formally accepted for publication once it meets all outstanding technical requirements.

Kind regards,

Colin Johnson, Ph.D.

Academic Editor

PLOS ONE
---

## [Editor Report · Acceptance letter]

9 Aug 2022

PONE-D-21-07892R3 

Uptake of premarital HIV testing and associated factors among women who had autonomous versus non autonomous marriage in Ethiopia: a nationwide study 

Dear Dr. Ahmed:

I'm pleased to inform you that your manuscript has been deemed suitable for publication in PLOS ONE. Congratulations! Your manuscript is now with our production department. 

Kind regards, 

on behalf of

Dr. Colin Johnson 

Academic Editor

PLOS ONE